# Improving Low Rank Coal Flotation Using a Mixture of Oleic Acid and Dodecane as Collector: A New Perspective on Synergetic Effect

**Maoyan An [1], Yinfei Liao [2,\*], Yijun Cao [2], Xiaodong Hao [3] and Longfei Ma [3]**

[1] Jiangsu Vocational Institute of Architectural Technology, Xuzhou 221116, China; anyan8601@126.com
[2] National Engineering Research Center of Coal Preparation and Purification, China University of Mining and Technology, Xuzhou 221116, China; caoyj@cumt.edu.cn
[3] School of Chemical Engineering and Technology, China University of Mining and Technology, Xuzhou 221116, China; kyzxb406@163.com (X.H.); cumt_mlf@163.com (L.M.)
\* Correspondence: liaoyinfei@cumt.edu.cn

**Abstract:** The mixed collector can improve low rank coal flotation efficiency, but its synergistic mechanism needs to be further explored. In this paper, oleic acid-dodecane (OA–D), oleic acid (OA), and dodecane (D) were employed to treat the low rank coal for revealing new synergistic mechanism of the mixed collector. First the surface free energy of the coal, the surface free energy of coal–water and coal–water–coal were calculated. Then wetting heat measurement, X-Ray Photoelectron Spectroscopy (XPS) and FTIR were used to analyze synergistic mechanism of the mixed collector in depth. The results showed that OA–D obtained a higher combustible recovery than using OA and D, respectively. The essence of synergetic mechanism of OA–D was that they formed a relatively ordered "supramolecular structure" on the low rank coal surface, especially there were hydrophobic and van der Waals forces between the oleic acid chain and the dodecane chain that can promote the formation of a continuous collector film.

**Keywords:** low rank coal; mixed collector; synergetic mechanism; supramolecular structure

## 1. Introduction

Coal, as a main energy source, makes an important contribution to the national economy with its consumption accounting for 58% of China's total energy consumption in 2018 [1]. However, the fine low rank coal often was discarded or stored in yard near the mine due to the difficulty of effective utilization, which caused severe environmental pollution and waste of resources. In 2017, the National Energy Administration of China clearly proposed the separation and utilization of low rank coal with late coal forming period, high volatile content, and low reactivity [2].

Froth flotation is a very useful method to facilitate the quality of coal. Unfortunately, fuel oil/kerosene, a conventional collector, is very inefficient to facilitate low rank coal flotation, and a considerable amount of collector is required to attain a satisfying recovery [3]. The high oxygen content, especially the abundance of polar groups (phenolic/alcoholic hydroxyl, carbonyl, carboxyl, ether, and methoxyl), results in the poor floatability of coal [4–7]. These hydrophilic functional groups weaken the surface hydrophobicity of low rank coal via absorbing water molecules to form hydration film [8]. Moreover, some weakly acidic groups (phenolic OH and COOH) ionize in solution making the coal surface negatively charged [9]. Low flotation efficiency, large consumption of collector, and high cost are the main problems that restrict the efficient utilization of fine low rank coal.

Electrokinetic properties and contact angle of coal–water are used to characterize the wettability of coal that is essential to the bubble–particle attachment and mineralization

process. Numbers of studies have been done to change the wettability of low rank coal. Heteropolar collectors, such as lubricating oil, vegetable oils, amine, carboxylic acid, $\alpha$-furanacrylic acid, and tetrahydrofuran ester series, were used to promote hydrophobicity of coal surface with the rapid development of a relevant interaction mechanism [10–16]. Jia et al. [14] pioneered the low rank/oxidized coal flotation with a series of tetrahydrofuran (THF) esters as collectors. It was revealed that a higher flotation efficiency can be obtained thanks to the hydrogen bonds between THF series and coal surface. Gui et al. [15] proposed that $\alpha$-furanacrylic acid as collectors interacting with the oxidized coal surface must be connected by water molecules in hydration film instead of directly interacting with the hydrophilic functional groups of the coal surface. The tendency of the collector to spread over the coal surface was due to the high interfacial interaction free energy between the collectors and the coal surface. In general, hydrophobic interaction, $\pi$-bonding, hydrogen bonding, and electrostatic force are major components of interaction forces of coal surface-heteropolar collectors [14,15,17]. For poor floatability of coal, it was accepted that the polar head of the collector interacted with its polar functional groups on the surface. Compared to the single heteropolar/nonpolar reagent, mixtures of heteropolar reagents and nonpolar materials, namely compound/mixed collectors, including oxidized diesel, biodiesel, waste vegetable oil, and polar/nonpolar mixture [18–21], have all been suggested to be more effective in facilitating the recovery of low rank/oxidized coal. It was indicated that the compositions of some collectors were more complex with polar groups, and the low rank/oxidized coal recovery was increased by the synergistic effect of different polar oil molecules [22]. Furthermore, surfactants have often been employed to combine with oil to promote floatability of low rank coal [13,23–31]. When surfactants were used as emulsifiers, oily collectors were dispersed into a large number of droplets [32]. The added numbers of oil droplets increased the collision probability between coal particles and oil droplets to assist flotation kinetics [33]. Furthermore, when surfactants were absorbed on the solid/water interface and solid/oil interface, the required energy of the oily collector spreading over the solid surface was reduced. Above methods markedly raise the low-rank coal flotation efficiency, but the related mechanism of collector-coal is not well understood [9,34].

Previous studies rarely considered the price and the source of reagents. The cheap and efficient mixed collector became our research object. Its flotation properties and related mechanism need to be explored. Herein, oleic acid-dodecane (OA–D), oleic acid (OA), and dodecane (D) was used as collectors and the surface free energy of low rank coal treated with different collectors at coal–gas, coal–water and coal–water–coal interfaces was calculated. The synergistic mechanism of OA–D absorbed on the coal surface was predicted by combining the law of wetting heat and the results of FTIR and X-Ray Photoelectron Spectroscopy (XPS). Furthermore, the improvement of the floatability of the low rank coal was investigated by flotation tests. Finally, the synergetic mechanism of OA–D in low rank coal flotation was proposed, that is, oleic acid and dodecane molecules formed relatively ordered "supramolecular structure" on the low rank coal surface.

## 2. Materials and Methods

### 2.1. Materials

A gravity concentrated sample, used as cleaned coal in theoretical analysis, was provided by Daliuta mine (Yu Lin city, Shaanxi Provence, China). This sample was first broken with a jaw crusher and then milled to a particle size < −0.5 mm (−74 μm accounting for 90%) in a laboratory ball mill (porcelain). Another sample, taken from products of raw coal bunker, was screened to collect particles smaller than 0.5 mm for the flotation test. The basic property analysis of the coal is listed in Table 1. Where volatile and H/C ratio of the coal samples used in this work was high, and the $C_{daf}$ of cleaned coal and fine raw coal

was 79.20% and 75.55% with 14.01% oxygen content and 15.35% oxygen content, respectively, which indicated that the samples was rich in oxygen-containing groups and belonged to the typical low rank coal.

S3500 laser particle sizer (MicotracInc, New Haven, VT, USA) was employed to measure the size distribution of the collected fine raw coal particles. Figure 1 showed that the percentage of size range of −50.26 μm and −99.66 μm were about 50% and 80%, respectively. Oleic acid (OA) and dodecane (D) were purchased from Aladdin and used as supplied. Oleic acid-dodecane (OA–D) used in this work was prepared by mixing OA and D with a mass ratio of 1:4. The frother, 2-octanol, was purchased from Sinopharm and used as supplied.

**Table 1.** Proximate and ultimate analysis of low rank coal samples.

| Samples | Proximate Analysis (%) | | | | Ultimate Analysis (%) | | | | |
|---|---|---|---|---|---|---|---|---|---|
| | $M_{ad}$ | $A_{ad}$ | $V_{daf}$ | $FC_{daf}$ | $C_{daf}$ | $H_{daf}$ | $O_{daf}$ | $N_{daf}$ | $S_{td}$ |
| Cleaned coal | 3.38 | 5.68 | 37.52 | 62.48 | 79.20 | 5.08 | 14.01 | 1.06 | 0.30 |
| Fine raw coal | 3.44 | 31.67 | 44.74 | 55.26 | 75.55 | 4.96 | 15.35 | 1.05 | 2.07 |

ad: air dry basis; daf: dry ash-free basis; t,d: total content, dry ash-free basis.

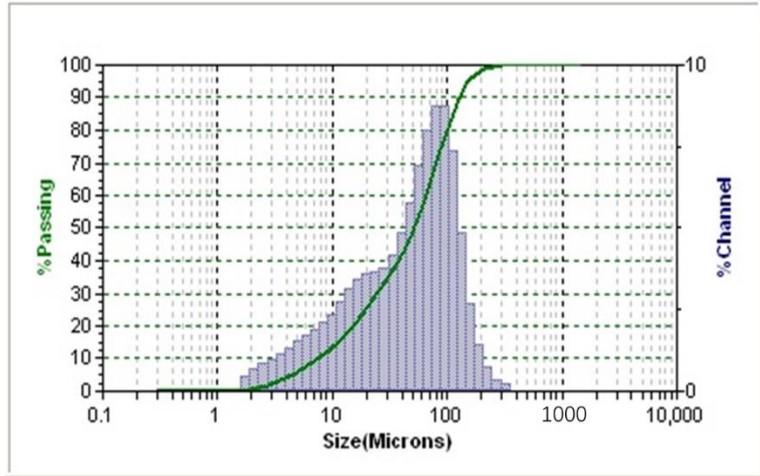

**Figure 1.** Particle size distribution of fine raw coal sample.

### 2.2. Materials

### 2.2.1. Theory

As we all know, the role of the collector is to enhance the hydrophobicity of the coal surface. This takes place via forming a nonpolar collector film on the coal surface, which lowers surface free energy to promote the mineralization of coal particles [35,36]. Therefore, calculation of surface/interface free energy of the coal before and after the collector's adsorption can predicted effectiveness of collectors.

On the basis of van Oss et al.'s method [37–39], the surface free energy of solids and liquids is made up of two components: Lifshitz–van der Waals ($\gamma^{LW}$) and Lewis acid–base ($\gamma^{AB}$).

$$\gamma = \gamma^{LW} + \gamma^{AB} \tag{1}$$

The nonpolar component ($\gamma^{LW}$) roots in dispersion, dipole–dipole type and dipole–induced-dipole intermolecular interactions [37]. The polar component ($\gamma^{AB}$) roots in complementary acid–base interactions, which can be written as the following formula:

$$\gamma^{AB} = 2(\gamma^+ \gamma^-)^{1/2} \tag{2}$$

where $\gamma^+$, the parameter of surface free energy, represents electron-acceptor intermolecular interactions (Lewis acid), and $\gamma^-$, the parameter deriving from electron-donor interactions (Lewis base). On the basis of van Oss et al.' s method, solid–liquid interfacial free energy $\gamma_{SL}$ is of the form [40]:

$$\gamma_{SL} = \gamma_S + \gamma_L - 2(\gamma_S^{LW} \gamma_L^{LW})^{1/2} - 2(\gamma_S^+ \gamma_L^-)^{1/2} - 2(\gamma_S^- \gamma_L^+)^{1/2} \tag{3}$$

where indexed S and L denote the solid and the liquid, respectively. When a liquid is placed on a solid, the solid–liquid phase and the surrounding steam are in equilibrium. So, well-known Young's equation can be expressed as:

$$\gamma_S - \gamma_{SL} = \gamma_L \cos\theta \tag{4}$$

where $\theta$ represents contact angle in the solid–liquid–air system. Introducing Equation (4) into Equation (3), Equation (5) can be obtained:

$$\gamma_L(1 + \cos\theta) = 2(\gamma_S^{LW} \gamma_L^{LW})^{1/2} + 2(\gamma_S^+ \gamma_L^-)^{1/2} + 2(\gamma_S^- \gamma_L^+)^{1/2} \tag{5}$$

From Equation (5), components of solid surface free energy, $\gamma_s^{LW}$, $\gamma_s^+$, and $\gamma_s^-$, can be calculated via measuring contact angles for three different liquids on the same solid surface, one nonpolar and two polar.

According to the Good–Girifalco–Fowkers method, the Gibbs free energy change of water and mineral surface per unit area can be expressed as:

$$\Delta G_{sw} = -W_{sw} = -(W_{sw}^{LW} + W_{sw}^{AB}) \tag{6}$$

where $W_{SW}^{LW}$ and $W_{SW}^{AB}$ are Lifshitz–van der Waals interaction and Lewis acid–base interaction, respectively. $W_{SW}^{LW}$ can be written in the form:

$$W_{sw}^{LW} = 2(\gamma_S^{LW} \gamma_W^{LW})^{1/2} \tag{7}$$

where $\gamma_S^{LW}$ and $\gamma_W^{LW}$ represents the Lifshitz–van der Waals interactions of coal and water, respectively. Lifshitz–van der Waals interaction has symmetry, and the hypothesis of Equation (8) is valid, while Lewis acid–base interaction is not applicable. The empirical expression assumed by van Oss et al. [36] can be used:

$$W_{sw}^{AB} = 2(\gamma_S^+ \gamma_W^-)^{1/2} + 2(\gamma_S^- \gamma_W^+)^{1/2} \tag{8}$$

where $\gamma_s^+$, $\gamma_w^+$, and $\gamma_s^-$, $\gamma_w^-$ are Lewis acid and Lewis base components of solid and water, respectively.

Introducing Equation (7) and Equation (8) into Equation (6), the change in the total Gibbs free energy of the solid–water adhesion is as follows:

$$\Delta G_{sw} = -W_{sw} = -(W_{sw}^{LW} + W_{sw}^{AB}) = -2((\gamma_S^{LW} \gamma_W^{LW})^{1/2} + (\gamma_S^+ \gamma_W^-)^{1/2} + (\gamma_S^- \gamma_W^+)^{1/2})) \tag{9}$$

In the flotation system, the coal was treated with collectors, and the interaction free energy between coal particles in solution can be done using following equation:

$$\Delta G_{SWS} = \Delta G_{SWS}^{LW} + \Delta G_{SWS}^{AB} + \Delta G_{SWS}^{EL} \tag{10}$$

where indexed W and S denote water and coal, respectively. LW, AB, and EL represent free energy of interactions rooting in Lifshitz–van der Waals, Lewis acid–base and electrostatic intermolecular interactions, respectively. Herein, we assume that $\Delta G_{SWS}^{EL} = 0$ for coal.

On the basis of van Oss et al.'s method [40], the $\Delta G_{SWS}^{LW}$ and $\Delta G_{SWS}^{AB}$ can be written as follows:

$$\Delta G_{SWS}^{LW} = -2\left[\left(\gamma_W^{LW}\right)^{1/2} - \left(\gamma_S^{LW}\right)^{1/2}\right]^2 \tag{11}$$

$$\Delta G_{SWS}^{AB} = -4\left[\left(\gamma_W^+\right)^{1/2} - \left(\gamma_S^+\right)^{1/2}\right]\left[\left(\gamma_W^-\right)^{1/2} - \left(\gamma_S^-\right)^{1/2}\right] \tag{12}$$

### 2.2.2. Wetting Contact Angle Measurements

Tensiometer K100 (Hamburg, Germany, KRUSS) was used to test the wetting behavior of the detection liquid on the surface of cleaned coal particles before and after interaction with reagents. In the test, $\alpha$-bromonaphthalene, formamide, and deionized water were employed as the probe liquid, and n-hexane was the reference liquid for the determination of the capillary constant of the Washburn tube. A mass of about 2 g of fine cleaned coal below 0.074 µm was used to fill a Washburn tube. After compacting, the height of the sample was about 2/3 that of the Washburn tube. The Washburn tube was attached to the  hook of a microbalance. A glass beaker with 30 mL liquid was moved slowly via a lifting platform until the distance from liquid level to Washburn tube bottom was about 2 mm. The test was started to get the wetting curve, and the wetting curve was fitted to obtain the slope, i.e., wetting contact angle θ.

### 2.2.3. Wetting Heat Measurements

A C80 calorimeter (Setaram, Caluire, France) was employed to record wetting heat flows between collectors (OA–D, OA, and D) and cleaned coal below 0.074 µm by the microcalorimetry method. At first, 8 mg of the coal sample and 2 mL of the collector were put into the bottom and upper parts of the cell, respectively. Then, the firmware of the cell was installed and put into the adiabatic cavity of the microcalorimeter. The aluminum foil film was pierced to make the solid–liquid mixture after the system was stable at 30 °C. Subsequently, wetting heat flow was recorded through Date Acquisition software, and the heat flow curve was manually integrated to obtain the wetting heat value.

### 2.2.4. FTIR Spectra Measurements

FTIR spectra were recorded using Fourier Transform Infrared Spectroscopy (Vertex 80v, Bruker, Leipzig, Germany). The sample and KBr solid powder were dried in a vacuum drying oven to eliminate the influence of water on the infrared test. Then, 20 mg of flotation cleaned coal was mixed with 2 g of KBr powder in an agate. After it was ground to further reduce the particle size and mix completely, the mixture was pressed into a thin plate under the pressure of 15 MPa for a duration of 1 min. FTIR spectra were obtained in the range of wavenumber from 4000 to 400 cm$^{-1}$ during 32 scans, with spectral resolution of 4 cm$^{-1}$.

### 2.2.5. X-Ray Photoelectron Spectroscopy (XPS) Measurements

XPS patterns were collected by an X-ray diffractometer (ESCALAB 250 Xi, Waltham , MA, America) for understanding the elements and content of cleaned coal after the adsorption of collectors. One gram of cleaned coal particles (−74 µm) and 0.05 g of the collector were placed into a breaker, then 100 mL DI (deionized) water was added with magnetic stirring at 800 rpm for 2 h, which was washed 3 times with DI water and vacuum-dried at 50 °C before measurement. The spectrum of the survey scan was recorded at the

pass energy of 100 eV with the step size of 1.00 eV. Before the data analysis, the binding energies were adopted by setting the C1s hydrocarbon peak at 284.8 eV as standard binding energy. The quantification and curve fitting of the spectra were determined with XPS peak fit software.

### 2.2.6. Flotation Tests

Batch flotation tests were performed in a 1.0 L XFD flotation cell with 60 g/L of pulp density. The flotation tests were conducted in a room temperature of 20 °C. For a typical test, 2-octanol used as a frother was added into the flotation system at 200 g/t of dosage, and the agitation speed of the flotation machine and the air flow rate was keep constant at 1800 rpm and 0.25 L/min, respectively. Firstly, the coal sample and water was conditioned in the flotation cell for 2 min. After that, the collector, D, OA, or OA–D, was added into pulp and conditioning process lasted for 2 min of further agitating. Then 2-octanol was then injected for conditioning another 30 s. Finally, the flotation was started to collect concentrate for 5 min after introducing air. Froth concentrates were collected, filtered, dried, weighed, and taken for ash determination and combustible recovery.

## 3. Results and Discussion

### *3.1. Surface/Interface Free Energy Analysis*

#### 3.1.1. Wettability of Coal Particles in the Presence of Collectors

The contact angles for water ($\theta_w$), $\alpha$-bromonaphthalene ($\theta_B$), and formamide ($\theta_F$) on the coal surface treated with D, OA, and OA–D are listed in Table 2. The results showed that the highest value of contact angle was observed for $\theta_w$ and the lowest for $\theta_B$. For water and formamide, the contact angles for coal treated with D, OA, and OA–D were higher than untreated coal particles, and their order was OA–D > OA > D. For $\alpha$-bromonaphthalene, the contact angles for coal treated with D, OA, and OA–D were lower than untreated coal, and the contact angles decreased in the order of D > OA > OA–D. It was demonstrated that the coal surface changed to more hydrophobic and homogeneous after treated with nonpolar/polar reagent, and the better performance was obtained by the combination of the two.

**Table 2.** The contact angles for water ($\boldsymbol{\theta_W}$) $\alpha$-bromonaphthalene ($\boldsymbol{\theta_B}$), and formamide ($\boldsymbol{\theta_F}$) on the coal surface treated with D, OA, and OA–D.

| Samples | $\boldsymbol{\theta_W}$ (°) | $\boldsymbol{\theta_B}$ (°) | $\boldsymbol{\theta_F}$ (°) |
|---|---|---|---|
| Fine Coal particles | 66.50 | 34.46 | 35.96 |
| D + Coal | 70.58 | 32.16 | 38.25 |
| OA + Coal | 81.10 | 29.46 | 48.15 |
| OA–D + Coal | 88.67 | 27.66 | 52.05 |

#### 3.1.2. Surface Free Energy of Coal Surface Treated with Different Reagents

The values used for calculations are presented in Table 3, which were all taken from literatures [41–43]. The calculated results of surface free energy components of coal/coal + reagent are listed in Table 4.

**Table 3.** Surface tension parameters of probe liquids at 20 °C.

| Liquids | Parameters (mN/m) | | | | | Polarity |
|---|---|---|---|---|---|---|
| | $\gamma_L$ | $\gamma_L^{LW}$ | $\gamma_L^+$ | $\gamma_L^-$ | $\gamma_L^{AB}$ | |
| Water | 72 8 | 21.8 | 25.5 | 25.5 | 51 | Polar |
| $\alpha$-Bromonaphthalene | 44.4 | 44.4 | 0 | 0 | 0 | Nonpolar |
| Formamide | 58 | 39 | 2.28 | 39 | 19 | Polar |

**Table 4.** Surface free energy components of coal calculated from Equation (5).

| Samples | $\gamma_S^{LW}$ (mJ/m²) | $\gamma_S^{+}$ (mJ/m²) | $\gamma_S^{-}$ (mJ/m²) | $\gamma_S^{AB}$ (mJ/m²) | $\gamma_S$ (mJ/m²) |
|---|---|---|---|---|---|
| Coal | 36.96 | 2.66 | 8.07 | 9.27 | 46.23 |
| D + Coal | 37.86 | 2.43 | 5.57 | 7.36 | 45.22 |
| OA + Coal | 38.85 | 1.35 | 1.99 | 3.28 | 42.13 |
| OA–D + Coal | 39.48 | 1.21 | 0.22 | 1.03 | 40.51 |

Many organic components in low rank coal surface contain a multitude of polar groups including -OH, -COOH, >C=O, and -CO-, which can form hydrogen bonds with water molecules, causing the surface free energy to rise. Moreover, the $\pi$ bond in the aromatic ring of the coal surface plays a greater role in polar interactions with absorbed collectors. The $\pi$ bond, >C=O, and -CO- groups are the origin of the Lewis base ($\gamma^{-}$), and the -COOH group is the origin of Lewis acid ($\gamma^{+}$). Meanwhile, -OH is both the origin of Lewis acid and Lewis base part of $\gamma_s^{AB}$ [36]. The purpose of introducing collectors into the system was to reduce the contribution of these polar components and the surface free energy of low rank coal.

Table 4 showed that the surface free energies of coal treated with collectors were lower than that of untreated coal. Both $\gamma_s^{LW}$ and $\gamma_s^{AB}$ values depended on the kind of collectors absorbed on the coal surface. The relative order of $\gamma_s^{LW}$ components was D< OA < OA–D, while $\gamma_s^{AB}$ components appeared in the opposite trend. No matter whether the collectors worked or not, Lifsbitz–van der Waals ($\gamma_s^{LW}$) of coal surface free energy, was greater than Lewis acid–base components ($\gamma_s^{AB}$), which demonstrated that the nonpolar character of coal was more remarkable than the polar character that determined the wettability and hydrophobicity of the coal surface.

The surface free energy of coal treated with OA–D was the minimum, 40.51 mJ/m², which was 4.71 and 1.62 mJ/m² lower than that treated with D and OA, respectively. This indicated that OA–D decreased the surface free energy of coal by collaborative adsorption with D and OA reducing the nonpolar component and polar component, respectively.

3.1.3. Interaction Free Energy on Coal–water Interface

The calculated values of the free energy change between coal particles and water were listed in Table 5. The values of interaction free energy between water and coal particle were all negative, which suggested that the interactions were spontaneous and the hydration films can be formed on the particle surface. The greater the negative value was, the stronger the coal surface hydrophilicity was, and the surface was easier to be wetted by water. The negative value of Gibbs free energy between water and coal treated with reagents were reduced, which meant that the hydrophilicity was decreased and the hydration film formed on the coal surface was thinner. The surface hydrophilicity sequence of the coal treated with reagents was: D + coal > OA + coal > OA–D + coal. OA–D can change Gibbs free energy to the greatest extent with the absolute value of interaction energy 74.52 mJ/m², which was 27.41, 22.52, and 9.67 mJ/m² lower than that treated with no collector, D, and OA, respectively. This was due to the fact that OA and D were together adsorbed on coal surface to induce the strongest hydrophobic surface and the thinnest hydration film. The above results were well in agreement with the results of the coal surface free energy.

**Table 5.** Interaction free energy of between water and coal surface.

| Samples | $\Delta G_{SW}^{LW}$ (mJ/m²) | $\Delta G_{SW}^{AB}$ (mJ/m²) | $\Delta G_{SW}$ (mJ/m²) |
|---|---|---|---|
| Coal | −56.77 | −45.16 | −101.93 |
| D + Coal | −57.46 | −39.58 | −97.04 |

| | | | |
|---|---|---|---|
| OA + Coal | −58.20 | −25.98 | −84.19 |
| OA−D + Coal | −58.67 | −15.85 | −74.52 |

### 3.1.4. Interaction Free Energy between Coal Particles in the Solution Phase

The free energy of interaction between coal particles in the solution phase was calculated and summarized in Table 6. The free energy between coal particles in water phase was negative, which suggested that there were attractive interactions between coal particles and they could spontaneously flocculate. In the presence of collectors, the increase in the absolute values of free energy resulted from the Lewis acid–base force. The Lewis acid–base free energy increased after the adsorption of collectors, indicating that the Lewis acid–base interactions was strengthened and was the main interaction between coal particles. The coal surface became more homogeneous after treated with collectors, so the interactions between coal particles and the possibility of agglomeration improved. These results may facilitate the appearance of coarse particles enhancing the flotation efficiency. The free energy absolute value of coal particles treated with OA–D was the highest (77.58 mJ/m$^2$), which can be attributed to that OA and D collaborative adsorbed on coal surface which created a more hydrophobic surface.

**Table 6.** Interaction free energy between coal particles in the solution phase.

| Samples | $\Delta G_{SWS}^{LW}$(m J/m$^2$) | $\Delta G_{SWS}^{AB}$(m J/m$^2$) | $\Delta G_{SWS}^{LW}$(m J/m$^2$) |
|---|---|---|---|
| Coal | −3.98 | −30.21 | −34.19 |
| D + Coal | −4.40 | −37.56 | −41.96 |
| OA + Coal | −4.89 | −56.59 | −61.48 |
| OA−D + Coal | −5.21 | −72.37 | −77.58 |

### 3.2. Wetting Heat Analysis

In the flotation system, the collectors replace water molecules on the coal surface which can change wettability of coal surface. The energy released in this process can reflect the intensity of collector's adsorption. Figure 2 shows the heat flow curves between coal and collectors. No peak was observed for the heat flow of D which indicated that the interaction between D and coal was weak. The peaks for heat flows of OA and OA–D were observed, indicating that absorption strength between D/OA–D and coal increased. The related wetting heat is shown in Figure 3. The negative values of the wetting heat revealed that the adsorption process was exothermic and spontaneous. OA–D had the largest absolute value of wetting heat (207.14 g/J), which was about 2 times that of OA and 4 times that of D. This was mainly due to the fact that OA–D wetted the coal surface easier with strong forces including hydrogen bond and van der Waals between OA–D and coal surface. It can be predicted that OA–D was a more effective collector than OA/D due to its superior hydrophobic modification performance.

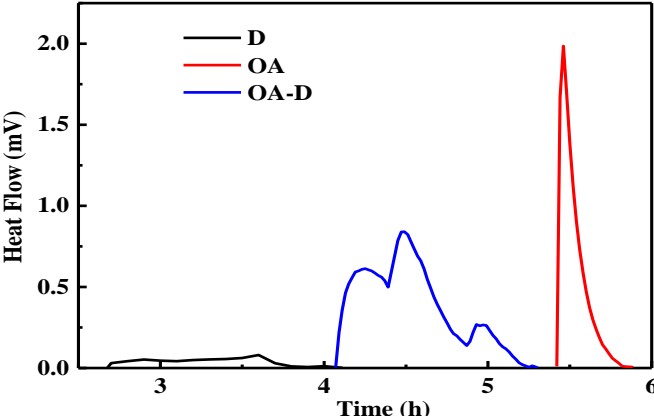

**Figure 2.** Compared heat flows of coal sample wetted by D, OA, and OA–D.

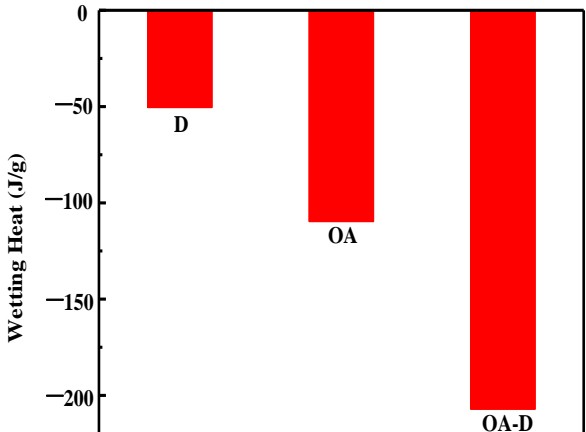

**Figure 3.** Wetting heat of coal sample wetted by different collectors.

### 3.3. FTIR Analysis

FTIR spectra of low rank coal treated with different collectors are shown in Figure 4. Whether coal was absorbed by single or combined collector, no new absorption peaks and no shift (shift > 4 cm$^{-1}$) of peaks appeared in the FTIR spectra of flotation concentrate, which gave an indication of there was physical adsorption between coal and collectors.

The relative absorbance of peaks at 3427, 2913, 1692, 1597, 1432, 1370, and 1100 cm$^{-1}$ attributing to organic constituents in coal were strengthened after the adsorption of collectors, which can be caused by the increase of the content of organic components. The main reason was that there were many polar groups in the organic components of the low rank coal, and these organic components were enriched during the flotation process. Meanwhile, characteristic peaks at 1033, 1011, 915, 539, and 471 cm$^{-1}$ relating to inorganic minerals declined significantly, which may be explained by the decline of ash and sulfur.

The amplitude intensity of the stretching vibration peak of OH at 3427 cm$^{-1}$ (assigned to alcohol or phenol) was in the order of D < OA < OA–D, indicating that the collecting ability of these three collectors also increased in turn. The peaks at 1033, 1011, 915, 539, and 471 cm$^{-1}$, relating to inorganic minerals, were reduced as follows: D > OA–D > OA. OA–D, a relatively effective collector in low rank coal flotation, had better collecting capability than a single reagent (OA/D), but its selection ability was in the middle. It integrated the collecting capability of OA and the selection ability of D.

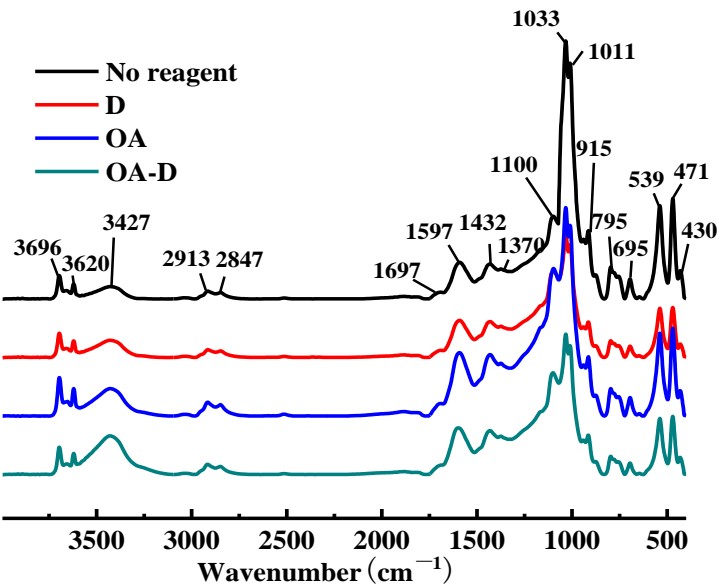

**Figure 4.** FTIR spectrums of low rank coal treated with different reagents.

### 3.4. XPS Analysis

In Figure 5, C1s peaks of coal surface were enhanced after adsorption of collectors, because some polar groups of the coal surface were covered and the hydrophobicity of coal surface was enhanced. C1s peak of the coal after the adsorption of OA was stronger than that of D, which indicated that OA had a strong modification effect on low rank coal surface. Compared to single collector, C1s peak of the coal after adsorbed OA–D was the strongest due to the synergistic effect of OA and D. Table 7 listed the contents of carbon and oxygen on low rank coal surface treated with collectors. As can be observed, for all the collectors, the carbon contents of low rank coal surface increased, but the oxygen contents decreased. It was evident that an increase of 0.77~2.41% in carbon content were obtained in the presence of D, OA, and OA–D, which can be attributed to the fact that the hydrophobic sites of coal surface were covered by a nonpolar D and hydrophilic sites of coal surface were masked by heteropolar OA. However, whether D or OA, it cannot be absorbed on the hydrophobic and hydrophilic sites of coal surface at the same time to enhance hydrophobicity to a large extent. Therefore, the combination of OA and D as a collector can completely enhance hydrophobicity of coal surface.

The C1s spectra of low rank coal with and without adsorption of collectors and its peak fitting are shown in Figure 6, whereas its carbon contents are listed in Table 8. As can be seen in Figure 6, there are four forms of carbon that exit in the surface structure of coal: aromatic graphitized carbon/aliphatic carbon (284.8 eV, C-C/C-H), phenolic/ether carbon (285.8~286.1 eV, C-O), carboxide (287.6~287.9 eV, C=O), and carboxyl (289.2~290.0 eV, COOH) [44]. Table 8 demonstrates that the major combination form of the carbon structure was C-C/C-H, accounting for 70.61%. The contents of C-O, C=O, and COOH were 19.90%, 5.35%, and 4.14%, respectively. With the adsorption of collectors, the order of the contents of C-C/C-H was OA–D > OA> D. When OA–D was used as a collector, the contents of C-C/C-H increased by 3.92% and the contents of C-O, C=O, and COOH fell by 3.3%, 0.42%, and 0.19%, respectively. However, the contents of C-C/C-H were only 1.96% and 3.27% higher than those of raw coal in the case of D and OA, respectively. Therefore, it can be concluded that the polar head of OA, containing COOH that was the hydrogen bond donor and acceptor, and can form a hydrogen bond with the oxygen-containing groups of the coal surface. The polar head of OA was mainly absorbed on the C-O group of coal surface and the nonpolar head extended outward, which facilitated D to absorb on

the hydrophobic area of coal surface or the area modified by OA. These findings once again demonstrated the synergistic effect between OA and D.

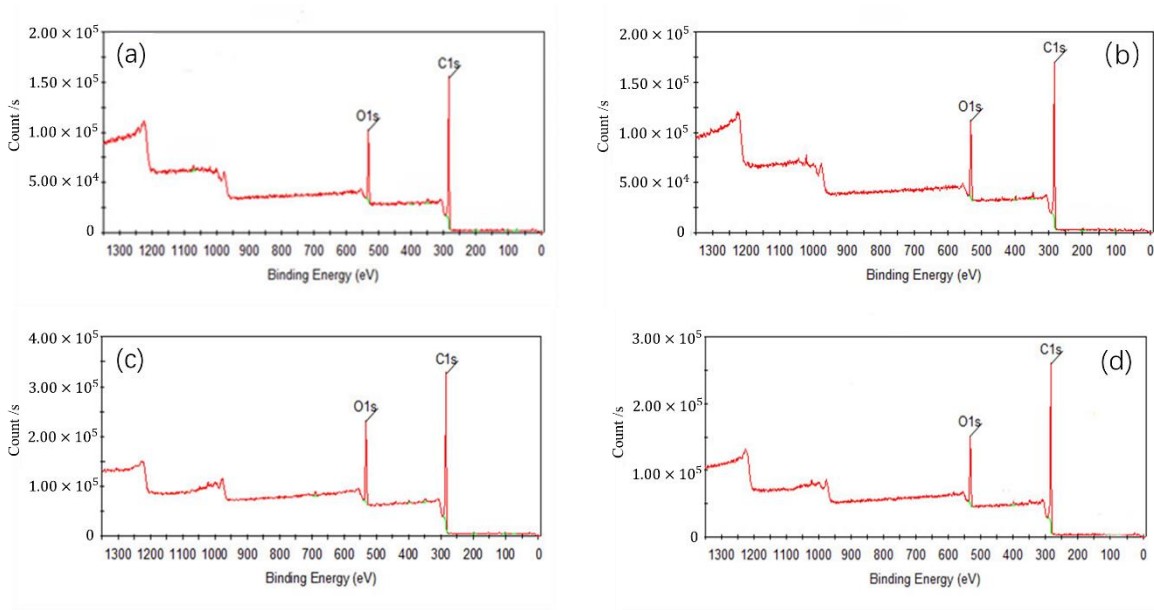

**Figure 5.** X-Ray Photoelectron Spectroscopy (XPS) wide energy spectra of coal surface treated with: (**a**) no collector, (**b**) D, (**c**) OA, and (**d**) OA–D.

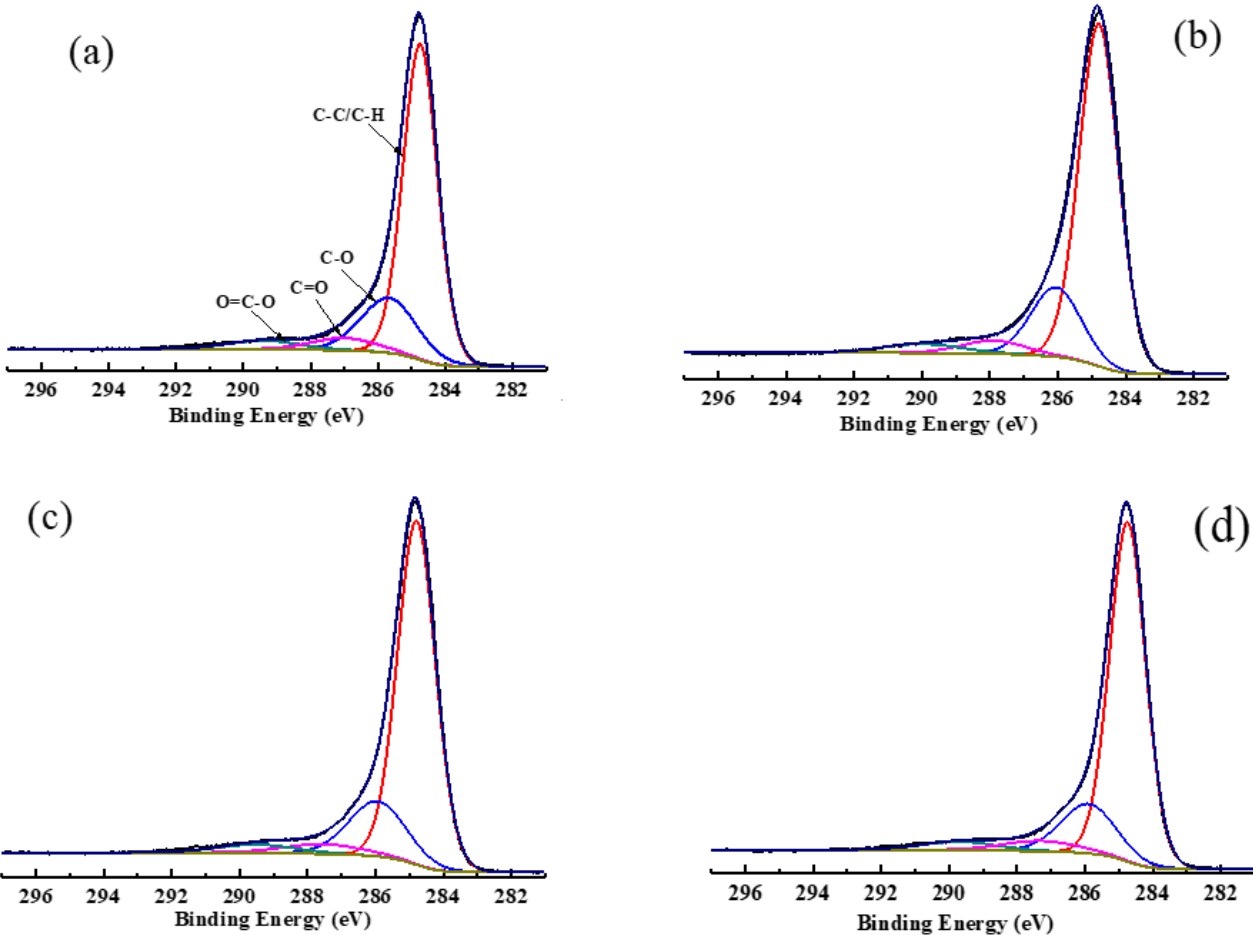

**Figure 6.** The C1s fitting spectra of low rank coal surface treated with: (**a**) no collector, (**b**) D (**c**) OA, and (**d**) OA–D.

**Table 7.** Carbon and oxygen contents on surface of the low rank coal treated with collectors (criterion: the total content of carbon and oxygen is 100%).

| Samples | $C_{1s}$ (%) | $O_{1s}$ (%) |
|---------|----------|----------|
| Coal | 81.71 | 18.29 |
| D | 82.48 | 17.52 |
| OA | 83.17 | 16.83 |
| OA–D | 84.12 | 15.88 |

**Table 8.** Contents of functional groups on the low rank coal surface after the adsorption of reagents.

| Samples | C-C,C-H (%) | C-O (%) | C=O (%) | COOH (%) |
|---------|-------------|---------|---------|----------|
| Coal | 70.61 | 19.90 | 5.35 | 4.14 |
| D + coal | 72.57 | 18.84 | 4.61 | 3.98 |
| OA + coal | 73.88 | 17.87 | 4.47 | 3.78 |
| OA–D + coal | 74.53 | 16.60 | 4.93 | 3.95 |

*3.5. Flotation Results*

The effect of collector dosage on combustible recovery and concentrate ash content is shown in Figures 7 and 8, respectively. In Figure 7, the combustible recovery increased with the increase of collector dosage. When the collector dosage was more than 3 kg/t, the recovery remained basically unchanged. At the collector dosage 3 kg/t of each collector, the order of combustible recovery under each collector was OA–D > OA > D. The combustible recovery was 89.45% with OA–D, which was 48.86% and 25.77% higher than with D and OA alone. As collector dosage increased, concentrated ash content decreased using D as a collector, while it increased in the case of D/OA–D. At the same collector dosage, OA had the highest concentrated ash content, and the lowest ash content was achieved by D, with OA–D in the middle. This might be due to OA bound with oxygen-containing groups of coal surface as well as gangue mineral. Therefore, the top flotation performance was obtained at OA–D dosage of 3 kg/t, with the highest combustible recovery and an acceptable concentrate ash content. These results were well consistent with the above conclusion about the synergistic effect of OA and D for low rank coal flotation.

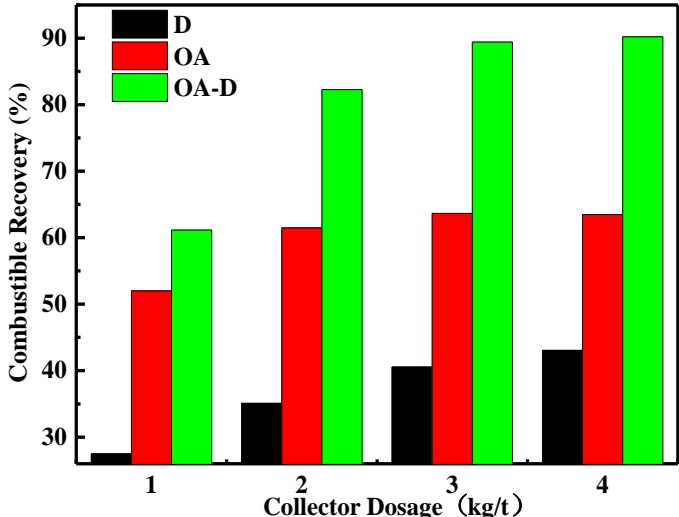

**Figure 7.** Effect of the different collector type on combustible recovery.

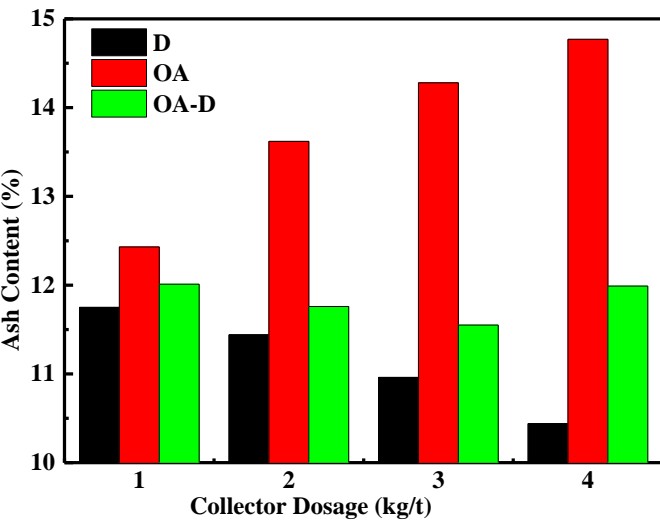

**Figure 8.** Effect of the different collector dosage on ash content.

### 3.6. Synergetic Mechanism

Higher combustible recovery can be obtained using OA–D as a collector than a single collector. The thermodynamic properties of coal particles treated with OA–D became more hydrophobic with thinner hydration film. The results of wetting heat, FTIR and XPS revealed that OA–D was the most effective collector making the low rank coal easier to float.

Different from the viewpoints of former scholars [21,23,27,45], we proposed that there might be a relatively ordered "supramolecular structure" when OA–D molecules were absorbed on the low rank coal surface. Figure 9 shows the formation process of the supramolecular system. OA–D droplets and low rank coal particles acted as guest and host, respectively. When OA–D droplets collided with the low rank coal particles, the small droplets were deformed and reconstructed in the molecular arrangement. OA molecules and D molecules were respectively adsorbed on the polar and nonpolar regions of coal surface via weak interactions, such as hydrogen bonds, van der Waals forces, and hydrophobic forces. Furthermore, the hydrocarbon chains of OA and D can interact by van der Waals forces and hydrophobic forces. Herein, the assembled complex of OA–D molecules with stable structure called "supramolecular structure".

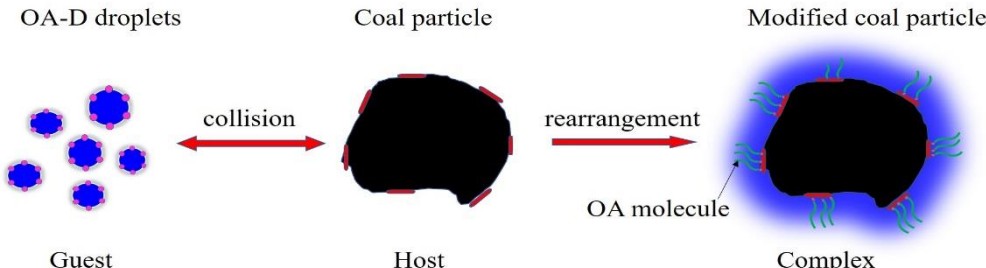

**Figure 9.** Forming process of the "supramolecular structure "of OA–D molecules on the low rank coal surface.

As seen in Figure 10, the low rank coal surface had hydrophobic sites including benzene, methyl and methylene et al., and hydrophilic sites including -OH, >C=O, -COOH, and -O- et al. When OA–D interacted with coal surface, the OA molecules recognized the hydrophilic sites and D molecules recognized hydrophobic sites. At the same time, the hydrocarbon chains between OA and D molecules interacted with each other, and the energy of the whole system decreased forming a relatively stable molecular layer. Either

OA or D interacted with the coal surface alone, there was just weak interaction between them. In case of OA–D, they matched together with the functional groups of low rank coal surface with considerable energy reduction, which was coincident with the results of wetting heat.

The adsorption of OA–D on low rank coal surface was of cooperativity, directivity, and selectivity. The additive and synergetic effect of each part can form strong intermolecular interaction. Although there was no chemical bond between OA–D molecules and functional groups of low rank coal surface, the total binding force between them was relatively strong. OA–D molecules can form compact molecular layers on the low rank coal surface, and the external of whole supramolecular body was all hydrocarbon chains with high hydrophobicity.

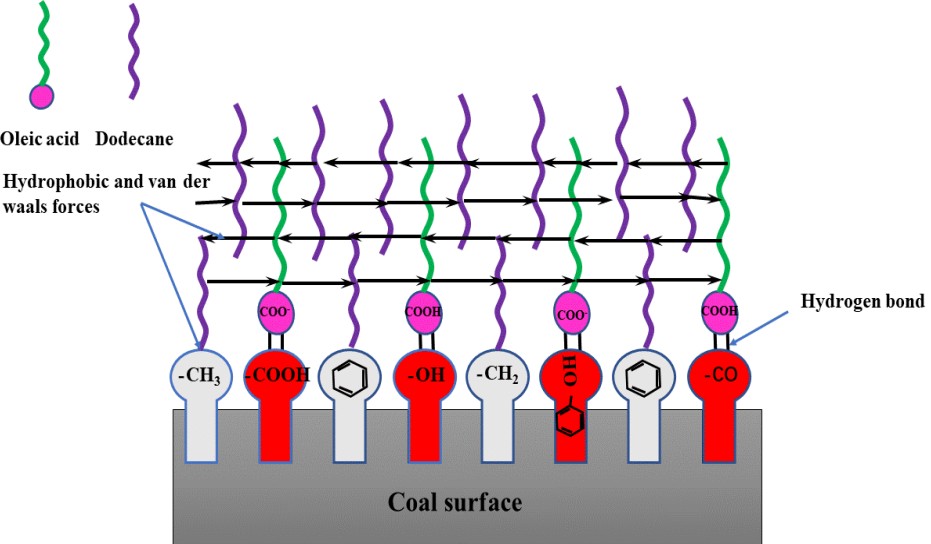

**Figure 10.** Schematic diagram of molecule interaction between OA–D molecules and the low rank coal surface in water phase.

## 4. Conclusions

In this work, a series of experiments were carried out to find a new perspective on the synergetic effect of oleic acid (OA) and dodecane (D) in improving low rank coal flotation. Thermodynamic results showed that coal particles treated with OA–D became more hydrophobic, had thinner hydrate film, and were more likely to condense, which was conducive to promote the floatability of coal. OA–D had the largest absolute value of wetting heat (207.14 g/J), which was about 2 times that of OA and 4 times that of D. According to the results of FTIR, OA–D had a better collecting capability than a single collector (OA or D), and its selection ability was in the middle. It integrated the collecting capability of OA and the selection ability of D. The polar head of OA, containing COOH that was the hydrogen bond donor and acceptor, can form hydrogen bonds with the oxygen-containing groups of the coal surface. The polar head of OA mainly absorbed on the C-O group of coal surface and the nonpolar head extended outward, which facilitated D to absorb on the hydrophobic area of coal surface or the area modified by OA. The optimal flotation results were obtained at an OA–D dosage of 3 kg/t with 89.54% combustible recovery and 11.54% concentrated ash content. Synergetic mechanism of OA–D molecules was that a relatively ordered "supramolecular structure" was formed on the coal surface. There were hydrophobic or van der Waals forces between oleic acid chain and dodecane chain, in addition to the hydrogen bonds, hydrophobic, or van der Waals forces between collector molecules and groups of the coal surface.

**Author Contributions:** Conceptualization, M.A. and Y.L.; methodology, Y.C.; investigation, X.H. and L.M.; resources, Y.L. and Y.C.; writing—original draft preparation, M.A.; writing—review and editing M.A. and Y.L. All authors have read and agreed to the published version of the manuscript.

**Funding:** This research was funded by the National Natural Science Foundation of China, grant number 52004283 and Special Doctoral Foundation of Jiangsu Vocational Institute of Architectural Technology, grant number JBJBZX20-10.

**Institutional Review Board Statement:** Not applicable.

**Informed Consent Statement:** Not applicable.

**Conflicts of Interest:** The authors declare no conflict of interest.

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
