# Peer review of "Improving Low Rank Coal Flotation Using a Mixture of Oleic Acid and Dodecane as Collector: A New Perspective on Synergetic Effect"

_processes, doi:10.3390/pr9030404_

Round 1

Reviewer 1 Report

315 At 3 kg/t collector

316 concentration, combustible recovery was as follows: OA-D > OA > D .

It is not clear, what was the collector dosage – 3 kg/t each or in sum? 

In the case 3 kg/t  is the individual dosage of D and OA, the sum is 6 kg/t. It is too much for flotation. If the sum is 3 kg/t, what is the ratio between D and OA?

3 kg/t  - is the collector dosage, not the collector concentration. Please, correct.

326      Collector concentratetion (kg/t) – Please, correct

Author Response

Review 1

Point 1: 315 At 3 kg/t collector

316 concentration, combustible recovery was as follows: OA-D > OA > D .

It is not clear, what was the collector dosage – 3 kg/t each or in sum?

Response: Thank you for the question. The results were given in the condition that each collector dosage wass 3 kg/t. We are sorry for the unclear statement. The sentence has been revised as follow:

At the collector dosage 3 kg/t of each collector, the order of combustible recovery under each collector was OA-D > OA > D.

Point 2: In the case 3 kg/t is the individual dosage of D and OA, the sum is 6 kg/t. It is too much for flotation. If the sum is 3 kg/t, what is the ratio between D and OA?

Response: The OA-D was applied 3 kg/t dosage in the flotation. The preparation of OA-D was given in section 2.1. The collector of OA and D was mixed in the mass ratio of 1:4.

Point 3: 3 kg/t - is the collector dosage, not the collector concentration. Please, correct.

326 Collector concentratetion (kg/t) – Please, correct

Response: Thanks for your correction. We have revised the ‘collector concentration’ into the ‘collector dosage’ in the manuscript.

Reviewer 2 Report

The authors used a combination of interesting methods to determine the flotation properties and related mechanism for the collector mixture.

A number of interesting procedures have been proposed that can be used to study flotation mechanisms in general. The results are presented clearly and legibly. 

Author Response

Response: We deeply apricate your review of our paper. Thanks very much for your positive comment.

Reviewer 3 Report

"Improving low rank coal flotation using a mixture of oleic acid and dodecane as collector: A new perspective on synergetic effect" is very interesting paper in separation processes. Some minor improvements can be made in text.

Line 171: Batch flotation tests were performed in a 1.0 L XFD flotation cell with 60 g/L of pulp density.  At which temperature? Influence of temperature was not considered in this study.Why?

Line 324 (Figure /): To correct " combusituble" at Y-axis with "combustible"

Line 326: Please to Change this title at X-axis: Collector concentratetion (kg/t) / correct " Collector concentration"

Line 342 (Figure 9): Is this model suitable for different coal particle size (nano, submicron, micron)? What are Limitations of this model?

Line 368, 369: The optimal flotation results were obtained at concentration of 3 kg/t with 89.54% combustible recovery. Can you give some Information About kinetics of Flotation in a Flotation cell?What is the limiting step of this Flotation?

General Questions:

  1. Is it possible to Transfer the obtained results from 1L to 10 L Flotation cell?
  2.  Why did you not offer SEM-Picture from supremolecular structure in order to confirm your model?
  3.  Is it possible to predict  kinetics of Flotation process  using of your proposed mechanism ?

Author Response

Point 1: Line 171: Batch flotation tests were performed in a 1.0 L XFD flotation cell with 60 g/L of pulp density.  At which temperature? Influence of temperature was not considered in this study.Why?

Response: Thank you for your question. We are sorry that we forget to mention the temperature condition of the experiment in our manuscript. The flotation tests were conducted at room temperature of 20 ℃, and the water/pulp was also 20 ℃.

The influence of temperature is very small but not negligible according to our previous experiment exploration. The reagent of dodecane (D) has a low melting point of -9.6 ℃, and it is insensitive to temperature. But oleic has a melting point of 13 ℃, and it is sensitive to temperature. However, the mixture of OA-D makes it less affected by the temperature because the melt temperature is decreased comparing a single collector. And this paper is focused on the mechanism of the synergistic effect of the collector rather than the influence of temperature on the collector performance. So, the influence of temperature was not further discussed in our paper.

The temperature condition has been added in the revised manuscript in section 2.2.6.

Point 2: Line 324 (Figure /): To correct " combusituble" at Y-axis with "combustible"

Response: Thanks for your reminder. The mistake has been revised.

Point 3: Line 326: Please to Change this title at X-axis: Collector concentratetion (kg/t) / correct " Collector concentration"

Response: Thanks for your reminder. The mistake has been revised.

Point 4: Line 342 (Figure 9): Is this model suitable for different coal particle size (nano, submicron, micron)? What are Limitations of this model?

Response: Thanks for your question. In coal flotation process, the coal particle was usually under the size of 0.5 mm. The coal size fraction of -50.26 μm and -99.66 μm account for 50% and 80% in our work, respectively (presented in Figure 1). The droplet size can change from 1 μm to 100 μm (according to our previous work ‘Enhancing low-rank coal flotation using mixed collector of dodecane and oleic acid: Effect of droplet dispersion and its interaction with coal particle, FUEL, 2020’). In Figure 9, the red mark on coal surface represents the polar region. The model fits the situation that the oil droplet was smaller than the coal particle. When the oil droplet was bigger than the particle, the size comparison between coal particle and oil droplets might be improper. However, the complex model (showed in the third picture in Figure 9) that formed by the oil and particle is still tenable because the interaction mechanism did not change, which still are the polar region adsorbed with OA molecular. No matter how small the coal particle is, the polar region will still exist in the surface, which means the OA molecular can interact with the coal. The model presented in Figure 9 was rather an illustration of the general case of the oil droplet-coal particle interaction than an accurate duplicate of the actual experimental phenomena.

This model can only demonstrate the organic interactions that happen between the regent and carbonaceous sites of coal. The influence brought by the small number of mineral impurities, the pores, and cracks that existing on the coal surface was limited by the illustration. The complex functional groups’ distribution on the coal surface was also limited by this model because the surface composition differs among coal with different ranks.

Point 5: Line 368, 369: The optimal flotation results were obtained at concentration of 3 kg/t with 89.54% combustible recovery. Can you give some Information About kinetics of Flotation in a Flotation cell? What is the limiting step of this Flotation?

Response: The flotation index including ash content and combustible material recovery rate were directly given in our paper. The experiments in our paper did not involve the kinetics of the flotation. In the flotation test, the concentrate yield will increase with time, and the increasing rate will decrease with time, which is consistent with the classics flotation kinetics model. All of the tests reached the end within the time of 5 min. The flotation kinetics was influenced by many factors, such as the pulp concentration, stirring speed, airflow rate. So, these parameters were explored in advance. The stirring speed, time, concentration, airflow rate are all in the optimal conditions in order to investigate the effect of the collector reagent.

General Questions:

Point 6: Is it possible to Transfer the obtained results from 1L to 10 L Flotation cell?

Response: The results can be duplicated on a larger scale experiment. The 1L cell is often applied in the laboratory flotation test and will be able to duplicate the results in the same type of laboratory flotation machine with 10L. If to conduct such experiments in 10L or even 1m3 cell in the pilot-scale test, the key point is that the flow field of the pulp and the air inflation need to be properly enlarged. In other words, the hydrodynamic environment of flotation cell should be consistent. According to our experience, for the same type of flotation machine with different volume produced by the same manufacturer, the reproducibility of flotation index is very good. So, it is possible to obtain a similar result in the 10L cell.

Point 7: Why did you not offer SEM-Picture from supremolecular structure in order to confirm your model?

Response: As far as we know, the SEM has a resolution of about 1 nm, and the TEM (Transmission Electron Microscope) has a resolution of about 0.2 nm. However, the current Electron Microscope can not directly observe the molecular structure of a chemical substance, Even the HRTEM (High Resolution Transmission Electron Microscope) can only observe the atoms column or lattice in the substance. So the supremolecular structure can not be confirmed by the SEM method. The STM (scanning tunnel microscope) method can observe the supramolecular structure but it requires a sample with smooth-surface, or regular lattice structure, and certain electroconductivity. The current STM is only applied to mineral materials. We think that the coal is not suitable for the STM measurement currently due to the heterogeneous structure of coal. So the supermolecular structure on coal surface can only be inferred by XPS or other measurements.

Point 8: Is it possible to predict  kinetics of Flotation process  using of your proposed mechanism ?

Response: This article has mainly discussed the mechanism in the aspect of surface chemistry, interface chemistry, and thermochemistry. The prediction of flotation kinetics is mainly based on hydrodynamic simulation, flotation kinetics experiments or models. The proposed mechanism in our paper can not be used to predict the flotation kinetics, but can be used to justify the flotation results based on the type of collector and coal sample. According to the proposed mechanism, the collector containing both alkane and oxygen-containing groups is more effective than the normal alkane-based collector in low-rank coal flotation. This prediction is also consistent with the previous studies on low-rank coal flotation (for example, in the paper Intensification mechanism of oxidized coal flotation by using oxygen-containing collector α-furanacrylic acid, Powder Technology, 2017).